# Safety Culture and Human Factors in Foreign Object Management in Surgery

**DOI:** 10.3390/healthcare13172167

**Published:** 2025-08-30

**Authors:** Sam Cromie, Alison Kay, Katie O’Byrne, Tess Traynor, David Smyth, Paul O’Connor, Dubhfeasa Slattery, Natalie Duda, Siobhan Corrigan

**Affiliations:** 1Centre for Innovative Human Systems (CIHS), School of Psychology, Trinity College Dublin, D02 PN40 Dublin, Ireland; kayam@tcd.ie (A.K.); dudan@tcd.ie (N.D.); 2University Hospital Waterford, Dunmore Rd, Ballynakill, X91 ER8E Waterford, Ireland; 3School of Medicine, University of Galway, 1 Distillery Road, H91 TK33 Galway, Ireland; paul.oconnor@universityofgalway.ie; 4Faculty of Medicine and Health Sciences, Royal College of Surgeons (RSCI) in Ireland, D02 PN40 Dublin, Ireland

**Keywords:** safety culture, foreign object retention, reporting culture, just culture, surgery, patient safety

## Abstract

Background: This paper examines the human and safety culture factors of the seemingly intractable problem of foreign object retention in surgery. Objectives: It reports selected findings of the FOR-RaM (Foreign Object Retention—Reduction and Mitigation) research project, which sought to discover and understand challenges to Foreign Object Management across surgical and maternity settings in Irish hospitals and to recommend changes to address these challenges. Methods: This paper presents the findings from surgical settings in one hospital site. A qualitative action research study was conducted with a wide range of stakeholders in the study hospital; the qualitative methods included 18 Semi-structured interviews with hospital staff, structured observations in surgical settings and Action Learning Sets to validate the data collected. Result: The results highlight a number of safety culture and human factors considerations which may facilitate or hinder Foreign Object Management, such as (individual and team) Goals, the Processes required for successful Foreign Object Management, Culture, Teamwork, Information Management, and Training.

## 1. Introduction

Retained Foreign Objects (RFOs) are recognised by healthcare services worldwide as serious adverse events. They are defined as “Any material or object related to an operative or invasive procedure that is unintentionally left inside a patient” [1]. The Joint Commission categorise them as sentinel events [1]; in the UK, they are on the Never Event List [2] and in Ireland are Serious Reportable Events [3].

While their occurrence is relatively rare, RFOs are potentially serious events with significant negative impacts on the patient, healthcare provider and the service involved [4]. Incidence estimates range between 1/1000 and 1/19,000 procedures [5]. Reliable comparative incidence figures are difficult to obtain since there are variable reporting practices [6]. Despite sustained focus by the patient safety community, there is little sign of the rates of RFOs declining [7].

The most fundamental element of the foreign object management process is the count. Typically, there are three to four counts [8]—the Preoperative, Initial or Baseline count before the surgery commences, the First Count (Closure of Cavity within a Cavity), the Second Count (Start of wound closure), Closing/Final count (Skin closure). Counts can also occur at any other time deemed necessary by a member of the team. The objective is to track and account for all objects entering the surgical field—primarily instruments and swabs. The count is performed by the scrub nurse and circulating nurse together, the latter recording the count; both are involved in counting the items, so both must visualise each item.

Counting is a very vulnerable activity—vulnerable to distraction and to lapses where objects are omitted from the count or double-counted. Good human factors practice is to seek to protect counting or measuring activities by carrying them out in quiet low distraction environments. The surgical count necessarily happens in far from ideal circumstances—in a team context where only two of the team are directly involved in the count and where the count may be delaying the progress of the operation.

Many RFO risk factors have been identified in the literature. Moffat-Bruce et al. [9] carried out a meta-analysis in which they found three high risk factors—(1) incorrect surgical count, (2) unexpected intraoperative factors, and (3) more than one surgical team, and four intermediate risk factors—(1) surgical count not performed, (2) more than one procedure, (3) long operations and estimated blood loss of more than 500 mL. Other studies have identified patient risk factors—such as high BMI [10,11]—and type of surgical procedure. Al-Qurayshi [11], for example, reports a higher risk only for abdominopelvic procedures. It is unclear if surgeons in different specialities have higher awareness of RFO risk. Even in specialities where the risk may appear to be low, RFO incidents still occur [12]. Specialities such as ENT and ophthalmology may not be typically considered high risk for RFOs, but incidents do still occur. Ophthalmology entails microscopic surgery, and the tiny needles (8.0 size) sometimes used can go missing. In this case, the surgeon would check the eye with the microscope instead of using X-ray and a magnet is rolled over the drapes and floor in the search for the object. In ENT surgeries, swabs may also go missing, as blood-soaked swabs are difficult to see in the throat, as in the case of tonsillectomy [12].

This identification of the RFO risk factors is informative but does not address the mechanisms of how RFOs occur nor provide insight for prevention or mitigation strategies. Moffat-Bruce’s risk factors comprise at least four different types of risk factors:Task-specific factors—more than one procedure, unexpected intraoperative factors, estimated blood loss;Operational consequences of such factors—long operations, more than one team;Deviations from foreign object management (FOM) procedure—count not performed;Alerts from the FOM procedure—incorrect surgical count.

It is likely that these risk factors are not independent; for example, multiple procedures may require more than one team—this may result in a greater likelihood of a count not being performed or the count being incorrect.

Hibbert et al. [13] presented a qualitative study of 31 Australian RFO root cause analysis reports. They focused on the three most common retained objects—surgical packs, drain tubes, and vascular devices. They identified a range of contributory factors in these reports, which resonate with other studies—lack of adherence to the count procedure, lack of standardisation of equipment, and communication failures. Steelman et al. [1] conducted a retrospective review of 308 RFO events reported to The Joint Commission. They identified 1156 factors contributing to 308 RFO events, most frequently in the categories of human factors, leadership, and communication.

These risk factors provide data on the “what, where and when” of RFOs but provide little insight into the how and why. This is where a sociotechnical systems analysis of the human and organisational factors, including cultural factors, is important. This has the potential to address the “how and why” of RFOs and pave the way for “how to” prevent or mitigate them.

“Poor communication between staff including failure to communicate suspicions and incompletely documented, non-standardized or incorrect counts” (Hibbert, 2019, p. 185 [13]) hint at potential cultural issues, but these are not explored. Why is there poor communication? Why are suspicions not communicated? Why are counts not fully documented? What are the influences on team and organisational culture in an operating theatre? Insight is needed into how these cultural factors interact with other human factors, such as task and team factors, in the foreign object management process.

Thiels et al. [14] reported the first prospective analysis of human factor elements in surgical never events, using the Human Factors Analysis and Classification System HFACS to classify the contributing human factors of 69 never events: 28% (19) of the events were retained foreign objects. Thiels et al.’s analysis encompasses four different surgical never events, including foreign object retention but unfortunately, does not separate out the specific sub-categories for RFO events, which limits the contribution of the paper to the understanding of the mechanisms underlying RFO.

A weakness of the HFACS system is that it categorises human factors as independent factors without providing any insight into how they may interact with each other in the foreign object management process. Steelman and Cullen’s (2011) study [15] starts to address this problem through a failure mode and effects analysis (FMEA) of retained surgical sponges. This method of analysis is well suited to elucidate both the system that is in place for managing RFO risk, which is often tacit knowledge for those experienced in the operating theatre, and for highlighting how it fails and where the vulnerabilities are. Steelman and Cullen first developed a map, based on observation, of the foreign object management process used in the study hospital. They then used this map as a tool to elicit from a focus group potential failure points in the six main steps in the process. Failures identified include five, which were associated with three different steps of sponge management: one person counting instead of two, not performing a count, not separating sponges, placing too many sponges into pockets and not counting a sponge. For each failure point, the group was then asked to identify potential causes of these failures. The most frequent causes identified were distraction, multi-tasking, not following procedure and time pressure.

Steelman and Cullen’s paper provides an insight into the dynamics of how different human factors challenges interact in the aetiology of an RFO event and provides a good foundation for the current paper.

This paper presents selected findings from the FOR-RaM project. It extends the preliminary finding previously reported [16]. As part of this project, the research team conducted a qualitative multi-method study in collaboration with one maternity hospital and one general hospital. However, this paper focuses on findings from only one of these sites—the general surgical hospital—and focuses on one aspect of the overall objectives which is to establish the ‘as is’ foreign object management process from a human factors and cultural perspective.

The full methodology for the project is described by Corrigan et al. [5] and Kay et al. [17]. The project adopted a Socio-Technical Systems (STS) approach to the challenge of foreign object retention. The STS approach understands safety and safety failures as emergent properties of the interaction and integration of humans, processes, information, technology, structures, and the external workplace environment [5,18]. STS incorporates both human factors and safety culture elements into its analysis. This project deployed an integrated evidence-based assessment methodology for social-technical modelling [17], the SCOPE Analysis Cube [19,20], and bow tie methodologies that were developed through a series of EU-funded projects. The SCOPE Analysis Cube focuses on the functionality and interactions of the current STS, which are key to understanding it more effectively, changing it to achieve better outcomes and an improved functioning for a safer future system. The elements of the SCOPE are Supply, Context, Organising, Process, and Effects.

The objective of this paper is to elucidate the human factors and safety culture of foreign object management in surgery that emerged from one of the hospital sites in this study.

## 2. Materials and Methods

### 2.1. Ethics

Full ethical approval was obtained across all hospital sites involved in the FOR-RaM project and from the Research Ethics Committee in the School of Psychology at Trinity College Dublin. All members of staff who took part in interviews, observations, and workshops underwent a briefing to make them aware of their rights as participants and to reassure them of confidentiality and anonymity. Participants were also briefed on the aims of the research and were provided with contact details and further information on the FOR-RaM project. All participants were asked for both their written and verbal consent to take part in the research before starting any research activities.

All patients who were observed in theatre were fully briefed about the research project and asked if they were happy for the researchers to observe their procedure. Patients were asked to provide written and verbal consent for researchers to observe their procedure. Patients were provided with a briefing sheet about the project, which contained contact details for the research team should they require more information or wish to withdraw from the research after the procedure was carried out.

### 2.2. Design

A qualitative multi-disciplinary action research study of foreign object management was implemented across two hospital sites, focusing on surgery in one and maternity services in the other. This study reports findings only from one hospital, with the focus on surgery.

The full study comprised four phases:Adapting SCOPE Analysis Cube [19,20] to this study;Developing process maps and a sociotechnical systems analysis;Examining risk in proposed change interventions;Selecting interventions and an implementation roadmap.

The focus of this study was on a qualitative understanding of surgical operations. Quantitative and wider organisational data were beyond the scope of this study—incident reports of RFOs were not studied, nor were audit data on compliance with the Surgical Safety Checklist.

The multiple phases and elements of this study are illustrated in Figure 1.

### 2.3. Interviews

Eighteen interviews were conducted across a range of roles in Hospital 1. The breadth of these roles is presented in Table 1:

Each of the interviews followed a validated semi-structured interview schedule and lasted between 15 and 60 min. All interviews were recorded where possible and fully transcribed. A thematic analysis was carried out. The transcripts from the semi-structured interviews were analysed separately by each member of the HF research team, and then inter-rater reliability was used to assess the level of agreement across the individual assessments.

### 2.4. Observations

Observations were completed across multiple specialities (ENT, General Surgery, Ophthalmology, Orthopaedics, Urology). Two observers were present during all observations-one with a clinical background and one with a background in human factors. An observational protocol was designed for use in theatre [17] the protocol included detailed observation of FOM activities and interactions, evaluating a range of performance shaping factors as well as a coordination demand analysis looking specifically at team factors. All observers underwent training in the use of the observation protocol.

### 2.5. Validation Workshops

An action learning set (ALS) was used as a means to validate the information obtained from interviews and workshops in Hospital 1. An ALS involved a group of hospital staff meeting to consider workplace problems and then generate a set of realistic actions to address them.

The action learning sets were presented to individual theatre staff during an interview, which lasted approximately 30 min. As with previous interviews, each staff member was given a briefing which included information about the project, intended length and running order of the interview, participant rights, confidentiality, anonymity, and right to withdraw. They were asked for both written and verbal consent—a copy of the information and consent forms can be found in the appendices.

Validation of the “To be” process maps: Following the briefing, participants were asked to evaluate the new “To be” process map. To do this, they were given a paper copy of the process map for the count and asked if there were any changes they would like to make. Participants were free to give verbal feedback or to provide written/drawn feedback on the paper copy.

Validation of datasets (high-level/interim findings): Participants were presented with cards from each individual category. They had been informed that these statements were taken from a previous round of interviews. They were asked to rank them in order of importance and to give the researchers their top three choices.

## 3. Results

These results arise from the analyses of interviews, workshops, and observations which took place in surgical settings in Hospital 1. Unfortunately, changes in personnel in Hospital 2 during COVID meant that it was not possible to validate and present the results with the same level of confidence. Results from the multiple methods presented above are combined for presentation.

Figure 2 provides an overview of the “As-is” STS analysis for Hospital 1 derived from the stakeholder interviews and observations, and the priority topics for attention (highlighted in bold) identified in validation workshops with stakeholders.

### 3.1. System Goals

Participants broadly agreed on the goals of the FOM system—patient safety, achieved through targeting zero incidents of RFOs, putting a high priority on FOM and taking shared responsibility for it. They felt that the process should be reactive in responding to issues that arise and proactive in anticipating and preventing them. Participants agreed that the “As-is” system falls down in over-emphasis on the reactive with little attention to the proactive. Participants also identified the definitive need for staff to become more proactive as a priority improvement to target with an intervention.

### 3.2. Process Elements

Teamwork, process, information, and culture are conceptualised as system elements that need to work together to ensure the system meets the above goals. Several key process elements were identified as key targets for intervention. While the Surgical Safety Checklist (SSC) is not part of the FOM process, the discipline of the checklist was considered important in that it promotes formal communication within the theatre, which is also essential to FOM. Formal communication in FOM itself is critical in the form of verbalised and acknowledged count outcomes. Should there be a discrepancy in the count, shared awareness and communication of missing items is crucial. Standardised practice across specialities would be ideal since personnel move from theatre to theatre, but this would be challenging to achieve since distinct practices exist across different specialities.

Two aspects of **information** were identified as needing attention in the system. There was no formal training in FOM so the process was passed on informally, which was seen as inefficient, and subject to procedural drift. There was also no practice of near-miss reporting; the database of events to inform process improvement was limited—to events categorised as incidents.

### 3.3. Teamwork Elements

Two teamwork elements were seen as needing to be addressed: (1) formidable personalities and (2) encouraging juniors to speak up. “Formidable personalities” is a generic phrase that was used to cover a variety of different behaviours and attitudes—impatience, a critical attitude towards and a tendency to blame other individuals and disciplines, unwillingness to teach, etc. Encouraging juniors to speak up is critical for FOM since it may often be a junior member of staff who is first to notice a problem, such as an incorrect count, and the effectiveness of the process depends on their being able to speak up. They could be junior in years, in the hierarchy of the theatre, or both. These elements are connected since it can be more difficult for a junior colleague to speak up with a “formidable personality” in the team.

### 3.4. Cultural Elements

Participants agreed that a culture of safe practice and putting the patient first is fundamental. Building on this, a culture that fosters good relations between staff is important to provide the environment where the teamwork process and information aspects of the system work effectively. Formidable personalities, identified as a problem in teamwork, were also identified as a problem of culture since it creates an atmosphere of lack of support.

Two aspects of culture were singled out as needing specific attention. Firstly, there is a need to formalise the understanding of what excellent behaviours, practices, and norms are. This echoes the comments under process and information, identifying the reliance on informal processes as a weakness of the system.

Secondly, participants said that it would be beneficial to understand “why” and “how” things happened. The culture may be stronger if richer details of incidents and near misses were shared to enable staff to understand the causes of the event rather than just its consequences.

## 4. Discussion

That retained foreign objects (RFOs) are an enduring problem in healthcare [7] may be a cause of bemusement in the public and a source of plotlines in medical dramas. To the layperson, it seems ridiculous that highly trained and highly paid surgeons using state-of-the-art equipment and techniques should make such an “elementary error”.

Yet from a human factors perspective, RFO’s status as an intractable problem is understandable. Many diverse objects are introduced into the surgical field in the context of a multidisciplinary team carrying out a complex technical and team task under time pressure with multiple potential complications.

### 4.1. The Foreign Object Management Process

The core process problem is that the foreign object management process is superimposed on the surgery process rather than integral to it. Foreign object management actions (counting, removing foreign objects, etc.) run parallel to surgical tasks, being neither cued by prior surgical sub-tasks nor required for subsequent tasks. This makes them more susceptible to a “lapse” type of error [21]. The challenge of FOM can also be compounded when items (such as surgical packing or tampons in obstetrics and gynaecology) are deliberately retained for a temporary period.

Foreign object management is an additional process superimposed on the core surgical task. Its function is a defensive one of preventing a relatively rare negative consequence; it is successful when nothing happens. This fact is key to the challenge of foreign object management.

Actions and interactions that are necessary will not be maintained by their immediate consequences in visibly progressing the task towards completion. They need to be maintained by a strong team and organisational culture that values them and reinforces their practice both tacitly and explicitly [22]. Otherwise, they may be subject to procedural drift and inconsistent application [23,24].

We examine these team and cultural factors in due course; however, first, let us unpack the elements of the foreign management process as articulated by the participants in this study. Five elements were articulated:Count outcome verbalised and acknowledged;Awareness and communication of missing items;Practice standardised across specialities;Surgical Safety Checklist [25]. (SSCL) sign-out formalised and fully completed; The WHO Surgical Safety Checklist is a 19-item checklist intended to decrease errors and adverse events, and increase teamwork and communication in surgery. It is used by a majority of surgical providers around the world.)SSCL promotes formal communication.

All five were designated as high priority in the validation workshops, signalling the value they put on the count.

**“Count outcome verbalised and acknowledged”** refers to the practice where, after the count, the scrub nurse states (normally) “count complete and correct” and this is acknowledged by the surgeon, typically by repeating the phrase. This small interaction is seen as significant since it formalises the count process as an integral part of the operation and extends the ownership of the process to the surgeon, and by implication to the whole team. Implicitly, it moves foreign object management from the sole responsibility of the nurses to the responsibility of the team [22]. There was some ambiguity about responsibility for foreign object management within the theatre. If asked who is responsible for the count, participants typically said the nurses were, but if asked who is responsible for foreign object management, they would say everyone is. In the concrete scenario where an object is missing, the whole theatre team would get involved—“get down on their hands and knees”—until the object is found.

It is in this context that **“Awareness and communication of missing items”** is significant. When an object goes missing—i.e., the count is incorrect—one of the nurses is required to state that the count is incorrect to escalate the problem from one of the counts (i.e., the responsibility of the nurses) to one of a potential retained foreign object (i.e., everyone’s responsibility). In the same way as verbalisation and acknowledgement of the count are strategically important when the count is correct, the communication of missing items is strategically important when the count is incorrect. Both, done well, strengthen the team culture and ownership of FOM; done badly, they fracture it.

That participants identified a **need for standardisation of FOM practice** across specialities emphasises the lack of this standardisation, not just across specialities but also between theatres and hospitals. This lack of standardisation means that every time there is a new configuration of a theatre team, there is at least a tacit negotiation of how the FOM process will happen. This implies a number of necessary questions to be understood by all in the theatre: Who will count? What will be counted? How will the count happen? Will the team be quiet for the count? Will the count outcome be verbalised and acknowledged? The **Surgical Safety Checklist (SSC)** only includes a record of FOM at the formal sign-out stage, where the surgeon and nurses responsible for the count confirm that the final count was correct. However, this is seen as significant in the FOM process since it promotes formal communication of this step. More broadly, where the SSC is practised, it provides an explicit formal social structure within which FOM can operate—it conveys that shared responsibility of safety is valued, empowering nurses to perform the FOM process with confidence.

Future research could benefit from complementing the qualitative approach in this study with gathering baseline and post-intervention quantitative data on SSC compliance and other quantifiable elements of the FOM process—adherence to “quiet for the count”, verbalisation, and acknowledgement of the count outcome. One of the challenges of this approach is that knowing compliance is being recorded is likely to change the behaviour being observed.

The hospital reports some changes in the audit procedures and SSC compliance since this study was completed [12]. Retrospective audits carried out monthly would have shown poor compliance with the sign-in and sign-out sections of the Safe Surgery Checklist (SSC) at the time of this study. However, since 2023, there has been a concerted effort to make improvements in compliance that included a change to documentation and multiple education sessions across all disciplines with a focus on the philosophy of psychological safety. A new biannual audit regime followed and includes 15% observational and 75% retrospective (maximum 50 audits depending on turnover). Compliance is reported to have improved significantly since the new interventions: sign-in is 90%, time-out is 100%, and sign-out (where the count is referenced) is 78%. The new results should be more accurate as the samples are taken across a one-week period.

A follow-up study to verify these changes and document whether they are complemented by qualitative changes in the process would be valuable.

### 4.2. The Role of Information

Two information elements were highlighted as priority: formal training in RFO prevention; 2. reporting of near misses. **Formal training in RFO** echoes the desire for standardisation of FOM practice. Not all members of surgical teams typically receive formal training in FOM—they pick it up in theatre through their internships and surgical residencies. The importance of **reporting near-misses** echoes the emphasis on being proactive in the goals. It is an acknowledgement that healthcare is not strong on near-miss reporting. Participants stated that when a count discrepancy is resolved—the object is found, or the counting error is identified—it is common practice for no report to be submitted. Their primary response was “this was not a ‘near-miss’” rather than “how can we learn from this?”

It was beyond the scope of this study to examine RFO incident reports in the hospital. This could be included in a future study; however, significant data protection challenges to such research exist in the Irish legal context.

### 4.3. Team Factors

It will be clear from the discussion of process factors above that the FOM process is a social process that explicitly involves some members of the theatre team but implicitly involves them all. The process can only thrive in a positive team culture. Two specific, and somewhat related, aspects of team culture were prioritised: 1. encouraging Juniors to speak up; 2. formidable personalities. **“Encouraging Juniors to speak up”** is an acknowledgement that it takes assertiveness to speak up in a theatre environment, especially if the message you are delivering is an unwelcome one, such as an incorrect account. It refers to juniors, acknowledging age/experience as factors that make it more difficult to be assertive; participants also reported cultural barriers in a multicultural workforce—speaking up is valued and encouraged more in some cultures than others. **“Formidable personalities”** are seen as making speaking up more difficult. The participants made it clear that the nursing staff would quickly learn—by experience and word of mouth—which surgeons created a positive team environment in their theatres, and which ones did not.

High power–distance relationships and strong hierarchies are characteristic of healthcare contexts, particularly surgery, and have long been highlighted as problematic for good human factors practice relevant to a range of patient safety issues, including RFOs [26]. Stevens et al. [27] conducted a systematic review of the literature on power within multidisciplinary health care team settings. The main impacts resulting from power imbalances they found were negative effects on team collaboration, decision-making, communication, and overall performance. All of these are relevant to FOM.

### 4.4. Cultural Factors and How They Influence Human Factors

It will be clear from the above discussion that the different elements of the FOM Socio-technical Systems do not operate in isolation—they mutually support or degrade each other. The organisational culture similarly shapes, and is shaped by, the FOM process, its use of information, the priority put on safety, and the local team context.

Two priority cultural factors were identified by the participants: 1. formalising excellent behaviours/norms/practice; 2. the benefits of knowing why and how things happened.

**Formalising excellent behaviours/norms/practice** echoes the desire for standardisation under process factors—a recognition that differing FOM practices should have to be navigated on a daily basis across theatres and teams. Formalising the best practice has a long tradition in safety management; the classic example being the extensive checklists used by pilots. While formalising and standardising FOM practices would not be a magic bullet—there are still many ways objects can conspire to be retained despite rigorous procedures—such formalising and standardisation should go some way to raising the collective vigilance in the team around RFOs.

One of the key reasons formal procedures and checklists fail is a lack of understanding of their purpose; as humans, we resist routines when we do not appreciate their benefit. Participants stated that it was **beneficial to know both “Why?” and “How?” things happen in FOM**. This echoes the desire for formal training in RFO prevention: the FOM process needs to be formalised and standardised, but team members also need to understand how and why foreign objects are retained and the role of the different procedural elements in protecting against these.

### 4.5. Contribution, Limitations, and Future Research Directions

The strength of this paper lies in its presentation of a rich picture of human and cultural factors behind foreign object management. The literature of FOM and patient safety generally would concur that the key elements highlighted—team dynamics, power distance, lack of standardisation, awareness, and communication—are common features of many healthcare settings. The central role of the specific foreign management process and how these human factors influence its integrity is a key contribution.

However, how these human factors play out could differ across different settings, and the FOM challenge would alter across different specialities. The maternity hospital that was hospital 2 in this study, for example, presented a specific challenge of FOM during transitions from delivery to the maternity suite. In this case, a speciality-specific adaptation of the FOM process was necessary. It is also possible that the results in this study were skewed by strong voices among the participants; this was mitigated by complementing workshops with interviews to ensure a diversity of voices. Observations, additionally, provided a third strand of data to help interpret and validate the data from interviews and workshops.

Keane et al. [28] report a successful quality improvement initiative targeted at RFOs. Their intervention was developed by a multidisciplinary team and focused on quiet time, minimising interruptions, and closed-loop communication during the final surgical count. They saw the number of RFO events dropping from 7 in 2021, prior to the intervention, to 0 in 2022, after it.

Such studies are promising and align well with the current study as they address the human factors we highlighted. It has been suggested that the success of interventions such as the one above makes the qualitative study of human factors in FOM, such as in our study, redundant. Why invest in understanding a problem when solutions are readily available, as was suggested by one of the original reviewers of our project proposal [29]? The answer is in the complexity of the systems addressed by quality interventions and the complexity of those interventions themselves. What works in one setting may not work in another setting or may eventually stop being effective in the original setting due to changes in context or procedural drift. Understanding the human and organisational factors underlying a patient safety challenge provides a stronger basis for designing interventions, implementing them, adjusting them, and translating them to new settings.

The multidisciplinary team approach to retained foreign objects implemented in the Mayo Clinic [30] is a very good model of an implementation program. It had surgical, nursing, and administrative leadership. It was based on a detailed review of FOR to identify organisational culture, institution, and operating room-specific failures. A thorough review of relevant institutional policies and procedures was conducted. The initiative was promoted in multiple and repetitive ways—all-staff conferences, training, simulation videos, daily education reminders, and in-room audits. Finally, in the monitoring phase, rapid leadership response teams were convened in response to any events. This provides an excellent model for implementing an FOM program in other institutions, but, in reality, other institutions may struggle to emulate this model if they do not have the economic resources, the leadership, the stable workforce, or the culture of improvement to work with.

## 5. Conclusions

RFOs continue to be a persistent problem in surgery. A number of technologies have been developed to help address the challenge, such as the use of Radio Frequency Identification (RFID) [31]. Artificial Intelligence holds promise for enhancing the detection of retained foreign objects [32]. The development of virtual assistants also holds promise for mitigating human limitations, such as concentration during instrument counts. Such technologies certainly could play a strong role in reducing RFOs. However, if they are to do so, they need to build on a strong foundation of effective human factors and safety culture. This paper helps to elucidate some of the key elements in strengthening this foundation. The applied challenge is to translate this knowledge into effective improvement at the theatre, hospital, and health system levels.

The potential for effective knowledge translation from this setting to other settings depends on the balancing of standardisation and customisation of the FOM process. Standardisation would yield the benefits highlighted above—staff from all disciplines would be able to readily adapt to FOM in new contexts. However, customisation would also be critical to ensure the process is relevant—this could require additions, such as a transfer protocol in a maternity setting, or adjustments, such as using a microscope rather than an X-ray in Ophthalmology contexts. A key third pillar of knowledge translation would be training—both FOM-specific training and HF training; this would be central to ensuring that what is transmitted is not just a rigid procedure, but a rich understanding of why the procedure is needed, how it should work, and how it depends on strong non-technical skills.

Lessons need to be learned from the challenges of rolling out the SSC. O’Connor et al. [33] outline implementation considerations: involvement of all members of the theatre team in the checklist process, demonstrated support for the checklist from senior personnel, ongoing education and training, and barriers to the implementation of the checklist to be addressed.

## Figures and Tables

**Figure 1 healthcare-13-02167-f001:**
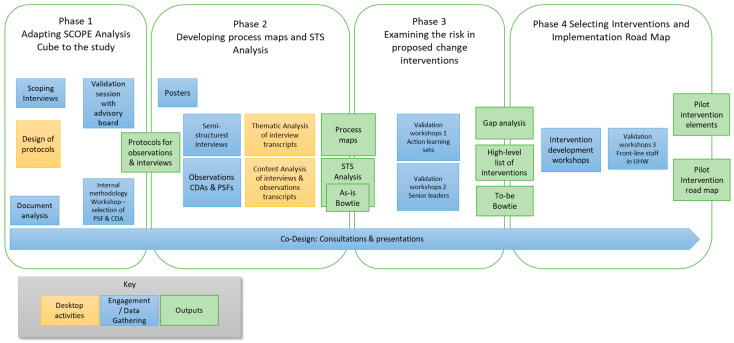
Research activities across the four phases. CDA = Coordination Demand Analysis, PSF = Performance Shaping Factors.

**Figure 2 healthcare-13-02167-f002:**
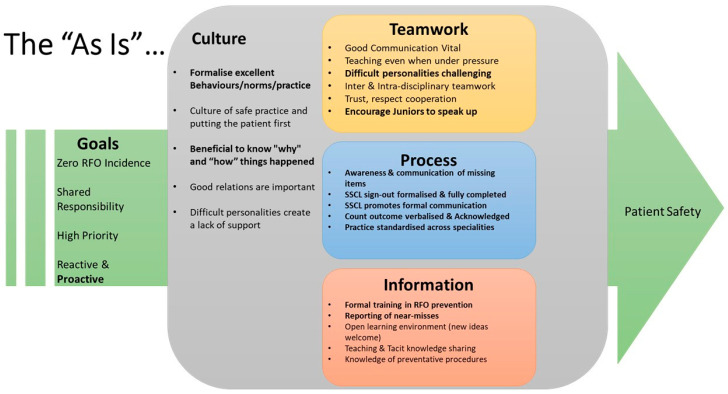
An overview of the critical human and organisational elements of the current Foreign Object Management process in Hospital 1. Items in bold were selected as priority topics for attention in validation workshops with stakeholders.

**Table 1 healthcare-13-02167-t001:** Interviewee roles.

Theatre Staff	Theatre Support Staff	Other Roles
All roles across Anaesthesia, ENT, General Surgery, Ophthalmology, Orthopaedics, Urology	Porter, Cleaning Services, Instrument Cleaning, Hospital Supplies	Clinical Nurse Managers, Hospital Directors, Members of Quality Improvement Team, Risk Management

## Data Availability

The datasets presented in this article are not readily available as individuals or teams are potentially identifiable from the data; consent for data collection was obtained on this basis.

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
