# Peer review of "Safety Culture and Human Factors in Foreign Object Management in Surgery"

_healthcare, 2025, doi:10.3390/healthcare13172167_

Round 1
Reviewer 1 Report
Comments and Suggestions for Authors
The article is engaging and explores the issue of retained foreign objects (RFOs), which, although rare relative to the number of surgical procedures performed, can result in serious complications and negatively impact the perception of the surgical team. While the topic has been widely studied, analyzing it through the lens of safety culture and human factors offers valuable insights and contributes additional knowledge for healthcare professionals.
The introduction is generally thorough; however, it could benefit from a brief mention of the types of procedures where RFOs are more commonly reported (e.g., abdominal, gynecological, and thoracic surgeries) as well as the patient profiles that carry a higher risk (e.g., individuals with a high body mass index). Surgical procedures should inherently alert the surgical team to heightened risk, as reliance on visual and tactile senses alone may not suffice—highlighting inherent human limitations.
The methodology is clearly described and appears to be appropriately applied. Nevertheless, it might be useful to gather baseline data regarding adherence to the surgical safety checklist in the examined settings. In some contexts, the checklist is treated as a mere formality, rather than as a meaningful expression of safety awareness and culture. Additionally, data on past RFO incidents could be valuable, as such events often leave a lasting impression on surgical teams.
It would also be interesting to know whether these incidents prompted internal audits. If so, this would demonstrate an organizational commitment to safety culture. In healthcare, audits following adverse events are a central element of clinical governance and risk management. Their implementation would reflect a positive effort toward continuous quality improvement and patient safety.
Specialties such as ENT and ophthalmology are typically not considered high-risk for RFOs. It would be insightful to assess whether professionals in these fields perceive the issue similarly to general surgeons.
Regarding hierarchical dynamics and relationships between senior and junior staff, it might be relevant to reference a foundational study that remains pertinent and could enrich the discussion.
Neil McIntyre and Karl Popper. The Critical Attitude In Medicine: The Need For A New Ethics. British Medical Journal (Clinical Research Edition) Vol. 287, No. 6409 (Dec. 24 - 31, 1983), pp. 1919-1923
Lastly, the discussion section should address the limitations and challenges encountered during the study. For instance, to what extent can the results from the validation workshop be generalized to other settings? Was there any participant bias or influence from dominant voices during group discussions that may have affected the outcomes? Furthermore, from an ethical perspective, what kind of feedback was obtained from patients involved in or informed about the study?
Reviewer 2 Report
Comments and Suggestions for Authors
The article addresses the issue of leaving foreign bodies in patients after surgical procedures and examines this problem through the lens of human factors, including cultural influences that contribute to such occurrences. The abstract is well-written, and the literature review highlights important studies on the topic. However, it would be beneficial to include information about the scale of the problem in specific (chosen) countries.
The research methodology is accurately described and appropriately chosen. The results are clearly discussed, but it is not specified which part of the research (interviews, workshops, or observations) produced these outcomes. The analysis of the results is thorough and engaging. Unfortunately, the conclusions are the weakest aspect of the article; they require enhancement and should align more closely with the article's purpose stated in line 151.
From my research perspective, it would also be valuable to explore the potential of using artificial intelligence to analyse surgical procedures, as virtual assistants could help mitigate the impact of human factors on adverse events.
Formal comments: The numbering of subsections, in particular, needs improvement across the board.
Reviewer 3 Report
Comments and Suggestions for Authors
Congratulations to the authors for the chosen research topics.
In nearly all surgical specialties, the risk of leaving a foreign object behind is a continuous concern for the surgical team, including both doctors and nurses. To address this issue, hospitals aim to implement protocols that maximize safety for both patients and the surgical team.
The authors carefully examined various aspects of this problem and conducted an elaborate study.
For scientific rigor, the authors must specify certain aspects that are not well understood in the materials and methods chapters.
The authors refer to a study conducted in two hospitals, but the findings pertain only to one. Incorporating data from maternity hospitals would have been beneficial, particularly in scenarios involving quick surgical procedures where intraoperative blood loss can be significant.
Another aspect to clarify is the study's limitations; the analysis took place at just one hospital, among other factors...
The conclusion chapter should shift the discussion of new technologies to the discussion chapter, which must be expanded.
Round 2
Reviewer 3 Report
Comments and Suggestions for Authors
Congratulations to the authors for the extensive changes made.
The subject is “difficult” to discuss and equally difficult to find solutions to reduce the incidence.
The authors' experience in this subject was useful in making the necessary changes for publication.
I am glad that the authors accepted the suggestions made and that they were used in a professional manner.
In my opinion, the scientific quality of the article has been greatly improved, and the article can be published.